# Isolation and Identification of Volatile Substances with Attractive Effects on *Wohlfahrtia magnifica* from Vagina of Bactrian Camel

**DOI:** 10.3390/vetsci9110637

**Published:** 2022-11-16

**Authors:** Jiaqi Xue, Dongdong Ai, Xiangjun Xu, Changmei Wang, Xinji Jiang, Tana Han, Demtu Er

**Affiliations:** 1College of Veterinary Medicine, Inner Mongolia Agricultural University, Key Laboratory of Clinical Diagnosis and Treatment Technology in Animal Disease, Ministry of Agriculture and Rural Affairs, Hohhot 010018, China; 2Alxa Left Banner Bayannorogon Comprehensive Administrative Law Enforcement Bureau, Bayannorogon 750300, China; 3Supply and Marketing Cooperative Union, Ejina Banner, Alxa League, Dalaihub 735400, China; 4Alxa Left Banner Centre of Animal Disease Prevention and Control, Alxa League, Bayanhot 750300, China; 5Comprehensive security and Technology Promotion Center of Dalaihub Town, Ejina Banner, Alxa League 735400, China

**Keywords:** Bactrian camel, vaginal myiasis, EAG, GC-EAD, biological prevention and control

## Abstract

**Simple Summary:**

Vaginal myiasis is one of the most serious parasitic diseases in Bactrian camels, and the disease has brought serious economic impacts to local herdsmen. Wohlfahrtia magnifica is the main pathogen causing hemorrhagic trauma and myiasis in the natural openings of humans and animals. The study showed that methylheptenone, 1-octen-3-ol, and propyl butyrate can all cause an antennae reaction of *Wohlfahrtia magnifica*. Moreover, except for propyl butyrate, the higher the concentrations of the other two compounds, the stronger the attractivity to the *Wohlfahrtia magnifica*, and the mixture of them at the ratio of 1:1 can enhance the attractivity. This study laid a foundation for biological control of vaginal myiasis in Bactrian camels.

**Abstract:**

Vaginal myiasis is one of the most serious parasitic diseases in Bactrian camels. At present, there are no reports on biological control measures of the disease. In this paper, the metabolomic analysis of vaginal secretions from susceptible and non-susceptible camels was performed by ACQUITY UPLC H-Class Ultra Performance Liquid Chromatograph. The results matched in 140 vaginal compounds. Methylheptenone, 1-octen-3-ol, and propyl butyrate and their mixtures were selected for gas chromatography-electroantennography (GC-EAD), electroantennography (EAG), behavioral experiments and trapping experiments of *Wohlfahrtia magnifica* (*W. magnifica*). Results showed that the *W. magnifica* had EAG responses to the three compounds, respectively. The EAG responses of female flies to different concentrations of methylheptenone were significantly different, but to the others had no significant difference, and there was no significant difference in the same compounds between the different sexes. Behavioral and trapping experiments showed that methylheptenone and 1-octen-3-ol have significant attraction to *W. magnifica*, but there was no significant difference to propyl butyrate. When methylheptenone and 1-octen-3-ol were mixed in different proportions, it was found that a mixture at the ratio of 1:1 and 0.5:1 had extremely significant and significant attraction, respectively, to both male and female *W. magnifica*. The study showed that, except for propyl butyrate, the higher the concentrations of the other two compounds, the stronger the attractivity to the *W. magnifica*, and a mixture at the ratio of 1:1 could enhance the attractivity to the *W. magnifica*.

## 1. Introduction

Bactrian camels are mainly distributed in Asia and surrounding cold desert areas, such as Mongolia, China, Kazakhstan, Russia and so on, and are an important material resource for people’s livelihoods. For a long time, the breeding industry of Bactrian camels has been affected by vaginal myiasis. The incidence of this disease is 20–30%, and the mortality rate is 2%. If the infected animals are not treated with deworming in time, the *W. magnifica* constantly produces maggots at the focus, the trauma of the body continues to worsen, the reproduction rate of sick camels declines, and the production of meat, milk and villi also declines, all of which cause serious economic impacts to the local herdsmen [1].

Some insects are good for humans, while others can cause serious diseases to humans and animals [2]. *W. magnifica* belongs to the order Diptera, family Sarcophagidae, genus Wohlfahrtia Brauer, which is widely distributed in Eurasia and other regions [3]. It is the main pathogen causing hemorrhagic trauma and myiasis in the natural openings of humans and animals [4,5]. It includes three metamorphic processes: larva, pupa. and adult. Larva within stage Ⅰ, Ⅱ and Ⅲ parasitize camel vaginas causing vaginal myiasis of Bactrian camels [6].

Volatile organic compounds (VOCs) are ubiquitous in insect communication, and are important chemical clues for insects to identify and locate food, find mates or avoid predators [7,8,9]. Smell, that is, the perception of chemicals from the air, is a key process of insect survival [10]. Antennae and maxillary tentacles are the main olfactory organs in insects, and many insects rely on them to find food and mates [11,12,13,14,15]. When the olfactory receptor receives the stimulation of volatile substances, the receptor converts the stimulation into electrical impulses and transmits them to the brain through neurons, thus producing different responses to different stimuli. In nature, insects’ sense of smell is usually stimulated by a mixture of chemical components. Different concentrations and proportions of these components can cause different behavioral responses, and at the same time, the ability to cause behavioral response is also different [16]. Studies have shown that herbivorous insects’ behavioral responses to the hosts’ mixed volatiles often exceed their responses to individual components [17]. With the recognition and accumulation of specific volatile compounds that have an impact on insect behavior, synthetic VOCs have been developed and applied to pest management [18].

Electroantennography (EAG) is a technique established by Schneider to study the olfactory pathway of insects [19]. EAG and gas chromatography-electroantennography (GC-EAD) are effective methods to study the identification and perception of volatile substances and pheromones in insects [20]. In recent years, GC-EAD has been widely used to screen chemical mixtures of plant volatile compounds and insect pheromones [21,22]. For example, by using GC-EAD, compounds that can be perceived through the olfactory receptors of *Drosophila melanogaster* can be identified, which lays a foundation for the development of new attractants [23].

At present, livestock myiasis caused by flies is mainly dewormed by drugs, and there are few methods for human intervention of flies in the environment. According to the statistics of our research group, in the Mongolian plateau, a female *W. magnifica* can lay and hatch about 100,000 adult flies in summer. Although pesticide deworming treatment can kill parasitic maggots, this has little impact on the *W. magnifica* in the environment, so it is difficult to control the disease by pesticide deworming alone. Biological control of pests is commonly used in agriculture and forestry, but rarely reports in the veterinary field. For pest management, it has been reported that, besides the use of pesticides, vector control can also be carried out through trapping experiments [24]. More and more studies have been devoted to the development of new trapping techniques for mosquitoes, including attempts to trap mosquitoes by using bait that attracts mosquitoes [25,26]. In some cases, the use of traps or a great number of recently developed trapping techniques have been proven to be effective and sustainable in reducing vector populations [27,28].

In this study, the vaginal secretions of susceptible and non-susceptible Bactrian camels with vaginal myiasis were identified and analyzed, and volatile substances with attractive effects on *W. magnifica* were screened, thereby finding a biological control method of vaginal myiasis, which in turn could reduce the contact opportunities between *W. magnifica* and Bactrian camels and reduce the incidence of the disease.

## 2. Materials and Methods

### 2.1. Liquid Mass Spectrometry Analysis of Vaginal Secretions from Bactrian Camels

The vaginal secretions of Bactrian camels were collected with sterile cotton swabs, where 10 samples, 10 samples, and 6 samples were collected from female camels that recovered after the disease, that were in a period of disease, and that did not suffer from the disease, and were recorded as group1, group2 and group3, respectively. The heads of cotton swabs with samples from each group were placed in a 5 mL centrifuge tube and marked, 1 mL ethyl acetate (Shanghai Maclin Biotechnology Co., LTD, Shanghai, China) was added, and the samples were fully dissolved by continuous oscillations of constant temperature ultrasonator in a water bath for 1 h. The solution was filtered into a sample bottle by inorganic filter and detected by Ultra Performance Liquid Chromatography.

Setting of test parameters of AQUITY UPLC H-Class Ultra Performance Liquid Chromatograph (Waters): ACQUITY UPLC C18 BEH (2.1 mm × 100 mm,1.7 µm, Waters) was selected as the chromatographic column; the column temperature and sample chamber temperature were constant at 40 °C and 10 °C, respectively; mobile phase A was water (0.1% methanoic acid); and mobile B was acetonitrile (0.1% methanoic acid). The test gradient and parameters are shown in the following table (Table 1 and Table 2).

### 2.2. GC-EAD of Methylheptenone, 1-octen-3-ol, and Propyl Butyrate

Methylheptenone (Shanghai Yuanye Biotechnology Co., LTD, Shanghai, China), 1-octen-3-ol (Shanghai Yuanye Biotechnology Co., LTD, Shanghai, China) and propyl butyrate (Shanghai Maclin Biotechnology Co., LTD, Shanghai, China) were prepared into 10^−2^ µg/µL samples with n-hexane (Shanghai Maclin Biotechnology Co., LTD, Shanghai, China), respectively. The antennae of the *W. magnifica* were cut off under a microscope, and the two ends of the antennae were connected to the corresponding electrodes using a capillary tube. The 1-day-old and 7-day-old *W. magnifica* were selected for the experiment. Each sample was repeated three times for male and female, and each antenna was used once.

Gas Chromatography (GC) (Agilent) conditions: the initial temperature was 120 °C, kept for 1 min, and then increased to 280 °C at a rate of 20 °C/min, and kept for 15 min; 5:10 split injection, each injection of 1 µL.

### 2.3. EAG of Methylheptenone, 1-octen-3-ol, and Propyl Butyrate

The three compounds were prepared into 10^−4^ µg/µL, 10^−3^ µg/µL, 10^−2^ µg/µL, 10^−1^ µg/µL, and 1 µg/µL samples with n-hexane, respectively.

Adult antennae of 5-day-old to 7-day-old *W. magnifica* were connected to an antenna potentiometer (SYNTECH). We took 10µL of the sample to be tested and dropped it on a 30 mm × 10 mm filter paper and placed it in a clean container. After the solvent was volatilized for 5 min, the antennae were stimulated. N-hexane was used as a blank control, and blank control experiments were conducted for each antenna before and after the test.

The stimulation time of each test was 0.2 s, the stimulation interval was 60 s, the stimulation airflow speed was 300 mL/min, and the continuous airflow was 1000 mL/min. Each sample was repeated six times for male and female. The sequence of EAG stimulation of each sample was blank control→10^−4^ µg/µL→10^−3^ µg/µL→10^−2^ µg/µL→10^−1^ µg/µL→1 µg/µL→ blank control.

The calculation formula of EAG relative reaction value is as follows:

EAG relative reaction value = (2 × Test the EAG value of the sample) ÷ (EAG value of solvent control before testing sample + EAG value of solvent control after testing sample).

The response data were analyzed and plotted by Graphpad Prism 8.0.1. One-Way ANOVA was used to analyze the EAG response of the same sex to different test samples and the EAG response of different sexes to the same test sample.

### 2.4. Behavioral Experiment of W. magnifica on Methylheptenone, 1-octen-3-ol, and Propyl Butyrate

The three compounds were prepared into 10^−4^ µg/µL, 10^−3^ µg/µL, 10^−2^ µg/µL, 10^−1^ µg/µL, and 1 µg/µL samples with n-hexane, respectively.

We selected 5-day-old to 7-day-old *W. magnifica* for behavioral experiments. Only one *W. magnifica* was placed into a Y-shaped tube each time, and the test time was 5 min. When it entered more than half of the side arm and remained there for at least 30 s, the results were recorded. If no choice was made within 5 min, the behavioral experiment for the *W. magnifica* was stopped. A group of 20 females and 20 males were selected, and the tests were repeated three times. N-hexane was used as blank control, and the experiment was carried out at room temperature.

Graphpad Prism was used for the *t*-test analysis of the results and plotted by Origin PRO.

### 2.5. Behavioral Experiment of W. magnifica on the Mixture of Methylheptenone and 1-octen-3-ol in Different Proportions

Two compounds were prepared according to the following table (Table 3), and all reagents were prepared 1 h before the experiment.

We selected 5-day-old to 7-day-old *W. magnifica* for behavioral experiments. Only one *W. magnifica* was placed into a Y-shaped tube each time, and the test time was 5 min. When it entered more than half of the side arm and remained there for at least 30 s, the results were recorded. If no choice was made within 5 min, the behavior experiment for the *W. magnifica* was stopped. A group of 20 females and 20 males were selected, and the tests were repeated three times. Distilled water was used as blank control, and the experiment was carried out at room temperature.

### 2.6. Trapping Experiment of W. magnifica on Different Concentrations of Methylheptenone, 1-octen-3-ol, and Their Mixtures in Different Proportions

Concentrations of 10^−2^ µg/µL, 10^−1^ µg/µL, and 1µg/µL methylheptenone, concentrations of 10^−1^ µg/µL and 1 µg/µL 1-octen-3-ol, and methylheptenone:1-octen-3-ol (1:1, 1:0.5) were selected for the trapping experiment on *W. magnifica*.

Our previous studies found that the behavioral response of *W. magnifica* mainly rely on olfactory sensors, so we did not apply light-avoidance treatment in this experiment. We took 2 mL of prepared reagent and put it into the bait basin of a fly catcher. Different concentrations of single products were compared with the same amount of n-hexane, while different proportions of mixtures were compared with the same amount of distilled water. The fly catcher was randomly hung in a fly cage. In this experiment, a hundred males and a hundred females were selected each time, and the experiment was repeated three times. Each experiment lasted for 12 h, and the data were recorded every 2 h.

The experiment data were analyzed and plotted by Graphpad Prism.

## 3. Results

### 3.1. Liquid Mass Spectrometry Analysis of Vaginal Secretions from Bactrian Camels

#### 3.1.1. UPLC-QTof/MS BPI Total Ion Chromatogram

UPLC-QTof/MS BPI total ion chromatograms of the three groups of samples were respectively drawn by ACQUITY UPLC H-Class Ultra Performance Liquid Chromatograph (Figure 1, Figure 2 and Figure 3). The results showed that the second group detected more abundant sample information.

#### 3.1.2. Structure Determination of the Main Components

The raw data were collected by ACQUITY UPLC H-Class Ultra Performance Liquid Chromatograph; the UNIFI was used for the qualitative analysis. Setting appropriate acquisition parameters and using the overall workflow of UNIFI software were combined with a self-built database of common vaginal compounds for data processing and structure matching. A total of 140 vaginal compounds of Bactrian camels were identified by the software, among which, 9 compounds with a high matching degree were identified (Figure 4).

#### 3.1.3. Multivariate Statistical Analysis-PCA Analysis

A PCA analysis was performed on the test data and a score plot was drawn (Figure 5). The PCA intuitively shows the separation of different groups of samples, that is, the obvious separation trend of groups of samples.

#### 3.1.4. Analysis Results of OPLS-DA

An OPLS discriminant analysis was performed on the test data of each of two groups of samples and an OPLS-DAS plot (“S” shape) was drawn to show the difference between the two groups. The compounds with the most significant difference between the two groups of samples were located on both sides of the “S” shape, which were potential markers with high research value [29].

There are significant differences in the score plots among all groups (Figure 6, Figure 7 and Figure 8), and the S-plot among all groups is shown in Figure 9, Figure 10 and Figure 11 (the green part is the same part; the blue and red parts at both ends are significant differences). The results showed that the contents of methylheptenone and 1-octen-3-ol were significantly increased in group 2 compared with group 1 and group 3, while propyl butyrate was significantly increased in group 3 compared with the other two groups.

### 3.2. GC-EAD Results of Methylheptenone, 1-octen-3-ol, and Propyl Butyrate

The results of GC-EAD showed that female and male *W. magnifica* at different ages responded to the three compounds (Figure 12, Figure 13, Figure 14, Figure 15, Figure 16 and Figure 17).

### 3.3. EAG Results of Methylheptenone, 1-octen-3-ol, and Propyl Butyrate

#### 3.3.1. EAG Results of Three Compounds in Different Concentrations for Male *W. magnifica*

There was no significant difference in the EAG relative response value of male *W. magnifica* to different concentrations of methylheptenone, 1-octen-3-ol, and propyl butyrate (*p* > 0.05) (Table 4, Figure 18, Figure 19 and Figure 20).

#### 3.3.2. EAG Results of three Compounds in Different Concentrations for Female *W. magnifica*

There was no significant difference in the EAG relative response values of female W. magnifica to different concentrations of 1-octen-3-ol and propyl butyrate (*p* > 0.05). The EAG relative response values of 1µg/µL methylheptenone were significantly higher than those of 10^−4^ µg/µL and 10^−3^ µg/µL (0.01 < *p* < 0.05), but the other concentrations were not significant (*p* > 0.05) (Table 5, Figure 21, Figure 22 and Figure 23).

#### 3.3.3. The Study of Comparative on EAG Responses of Different sexes of *W. magnifica* to the Same Compound

The results showed that there was no significant difference in the EAG relative reaction value of different sexes to methylheptenone, 1-octen-3-ol and propyl butyrate (*p* > 0.05) (Figure 24, Figure 25 and Figure 26).

### 3.4. Behavioral Experiment of W. magnifica on Methylheptenone, 1-octen-3-ol, and Propyl Butyrate

#### 3.4.1. Behavioral Response of Male *W. magnifica* to Different Concentrations of Methylheptenone, 1-octen-3-ol, and Propyl Butyrate

The experiment results showed that 10^−1^ µg/µL, 1 µg/µL methylheptenone and 1 µg/µL 1-octen-3-ol had extremely significant attraction to male *W. magnifica* (*p* < 0.01). 10^−2^ µg/µL methylheptenone and 10^−1^ µg/µL 1-octen-3-ol had significant attraction to them (0.01 < *p* < 0.05). Concentrations of 10^−3^ µg/µL, 10^−4^ µg/µL methylheptenone and 10^−2^ µg/µL, 10^−3^ µg/µL, 10^−4^ µg/µL 1-octen-3-ol had no attraction to them (*p* > 0.05). The low concentration of propyl butyrate has no significant attraction to the male *W. magnifica*, but compared with 10^−1^ µg/µL and 1 µg/µL propyl butyrate, the control group has significant attraction to them (Figure 27, Figure 28 and Figure 29).

#### 3.4.2. Behavioral Response of Female *W. magnifica* to Different Concentrations of Methylheptenone, 1-octen-3-ol, and Propyl Butyrate

The results showed that 10^−2^ µg/µL, 10^−1^ µg/µL, 1µg/µL methylheptenone and 10^−1^ µg/µL, 1 µg/µL 1-octen-3-ol had extremely significant attraction to female *W. magnifica* (*p* < 0.01). A concentration of 10^−2^ µg/µL 1-octen-3-ol had significant attraction on them (0.01 < *p* < 0.05). However, 10^−3^ µg/µL, 10^−4^ µg/µL methylheptenone and 10^−3^ µg/µL, 10^−4^ µg/µL 1-octen-3-ol had no significant attraction to them (*p* > 0.05). The low concentration of propyl butyrate has no significant attraction to the female *W. magnifica*, but compared with 10^−1^ µg/µL and 1 µg/µL propyl butyrate, the control group has significant attraction to them (Figure 30, Figure 31 and Figure 32).

#### 3.4.3. Comparative Study of the Behavioral Response of Different Sexes *W. magnifica* to the Same Compound

The results showed that 1 µg/µL methylheptenone had extremely significant attraction to female compared with male *W. magnifica* (*p* < 0.01). When the concentration was 10^−1^ µg/µL, it had significant attraction to female *W. magnifica* (0.01 < *p* < 0.05). The other concentrations of methylheptenone had no significant attraction to female or male *W. magnifica* (*p* > 0.05) (Figure 33).

The results of behavioral experiments of different sexes of *W. magnifica* to 1-octen-3-ol showed that when the concentration was 1 µg/µL, the attraction to females was significantly higher than that to males (0.01 < *p* < 0.05). The other concentrations of 1-octen-3-ol had no significant attraction to female or male *W. magnifica* (*p* > 0.05) (Figure 34).

The behavioral experiment of propyl butyrate showed that the attraction of propyl butyrate at five different concentrations were not significant for female and male *W. magnifica* (*p* > 0.05) (Figure 35).

#### 3.4.4. Behavioral Experiment of *W. magnifica* on the Mixture of Methylheptenone and 1-octen-3-ol in Different Proportions

The results showed that methylheptenone:1-octen-3-ol (1:1) had extremely significant attraction to both male and female *W. magnifica* (*p* < 0.01). Methylheptenone:1-octen-3-ol (0.5:1) had extremely significant attraction to female *W. magnifica* (*p* < 0.01). The other concentrations had no significant attraction to male or female *W. magnifica* (*p* > 0.05) (Figure 36).

#### 3.4.5. Trapping Experiment of *W. magnifica* on Different Concentrations of Methylheptenone, 1-octen-3-ol, and Their Mixtures in Different Proportions

The experiment results showed that compared with n-hexane, 1µg/µL methylheptenone and 1-octen-3-ol had extremely significant attraction to both male and female *W. magnifica* (*p* < 0.01). A concentration of 10^−1^ µg/µL methylheptenone had significant attraction to male *W. magnifica* (0.01 < *p* < 0.05), but no significant attraction to females. The other concentrations of methylheptenone and 1-octen-3-ol had no significant attraction to male or female *W. magnifica* (*p* > 0.05) (Table 6 and Table 7, Figure 37).

The results of trapping *W. magnifica* with the mixtures of methylheptenone and 1-octen-3-ol in different proportions showed that compared with distilled water, the mixture of methylheptenone and 1-octen-3-ol in the ratio of 1:1 had extremely significant attraction to both male and female *W. magnifica* (*p* < 0.01). The ratio of 0.5:1 had significant attraction to both male and female *W. magnifica* (0.01 < *p* < 0.05) (Table 8 and Table 9, Figure 38).

## 4. Discussion

At present, the treatment of vaginal myiasis of Bactrian camels rely on pesticide deworming, but this method can’t eradicate the disease, and would be cause environmental pollution. The results of liquid mass spectrometry showed that methylheptenone, 1-octen-3-ol, and propyl butyrate were the important components with attractive efficacy to *W. magnifica* in the vaginal secretions of Bactrian camels.

The GC-EAD and EAG results of *W. magnifica* to methylheptenone, 1-octen-3-ol, and propyl butyrate showed that there was a response to the three compounds at different ages of *W. magnifical*. Behavioral experiments and trapping experiments were carried out on W. magnifica with different concentrations of methylheptenone, 1-octen-3-ol, propyl butyrate, and mixtures of methylheptenone and 1-octen-3-ol in different proportions, the results showed that, except for propyl butyrate, the higher the concentration of the other two compounds, the more attractive to the *W. magnifica*, and the 1:1 ratio mixture of these could enhance the attraction to *W. magnifical*.

A host is indispensable for insect survival. Host volatiles play an important role in mate selection and reproduction for insects [30]. For example, the volatile of cow urine is attractive to *Musca autumnalis* [31]. Studies showed that lepidopteron insects rely on smell and plant volatiles for host selection, and they can identify and locate their hosts by sensing specific compounds from their hosts [32]. At present, GC-EAD and EAG are mainly used in the research of insects and their host plants and non-host plants.

The main function of GC-EAD is to detect whether there are substances that have relevant reactions to insect antennae from the crude extracts of host volatiles, and then using the separation and discrimination ability of GC to screen out the corresponding substances. EAG mainly reflects the reactions of insects to different compounds, so as to screen out the substances with stronger reactions. The results showed that the three compounds had EAG responses to both male and female *W. magnifica*, and the response of females was higher than males in general. Studies have found that ®-3-hydroxy-2-hexanone had EAG responses to both male and female *Diploschema rotundicolle* [33], similar to our results, while other studies have shown that the EAG response of male moths to (Z)-3-hexenol butyrate, (Z)-3-hexene-1-ol, and 3,7-dimethyl-1,3,7-octantriene was significantly higher than that of female moths [34]. The insect species, sex, sensitivity to compounds, and the concentration of compounds affects the EAG response. For example, compared with *Apis mellifera*, the EAG response of *Apis cerana* to queen mandibular pheromone is always weaker [35]; 1 µg of compound can cause a higher EAG response of *Triatoma dimidiate complex* [36]. In addition, the response of antennae to volatiles varies with the time of day because the olfactory response and the pulse-tracking ability of antennae follow the circadian rhythm of insects [37].

In addition to the use of pesticides, using insect pheromones and volatile compounds related to food to trap insect pests has become an effective pest control method. At present, there is no report on biological control methods of *W. magnifica*, but some studies have found that using human-odor extracts to attract adult female mosquitoes into traps can reduce the survival rate of mosquitoes [38]. Kaidi Cui found that 1-octen-3-ol is attractive to blood-sucking insects, and can be used as an attractant of mosquitoes [39]; in addition, it can inhibit *Drosophila melanogaster* and *Rhynchophorus ferrugineus* from ovipositing [40,41]. Many studies have found that mixing host volatiles can change the trapping rate [42]. For example, compared with 1-octen-3-ol alone, mixed use of it with CO_2_ can improve its trapping rate [43,44]; vanillin and DEET used at the same time can enhance the repellency of DEET to mosquitoes [45]; and a synthetic terpenoid mixture has strong attraction to houseflies [46]. Thus, adding particular combinations of attractants to traps can improve the trapping rate of insects [47]. Many insects detect their specific proportion of key volatiles to find their hosts; therefore, even a few small changes in proportions may increase or decrease the attraction of hosts to insects [48].

## 5. Conclusions

Our study showed that methylheptenone, 1-octen-3-ol, and propyl butyrate can cause an antennae reaction of *W. magnifica*. Different concentrations of methylheptenone, 1-octen-3-ol, and propyl butyrate are attractive to *W. magnifica*, and the mixture of methylheptenone and 1-octen-3-ol at the ratio of 1:1 can attract more *W. magnifica*. This experiment laid a foundation for biological control of vaginal myiasis in Bactrian camels.

## Figures and Tables

**Figure 1 vetsci-09-00637-f001:**
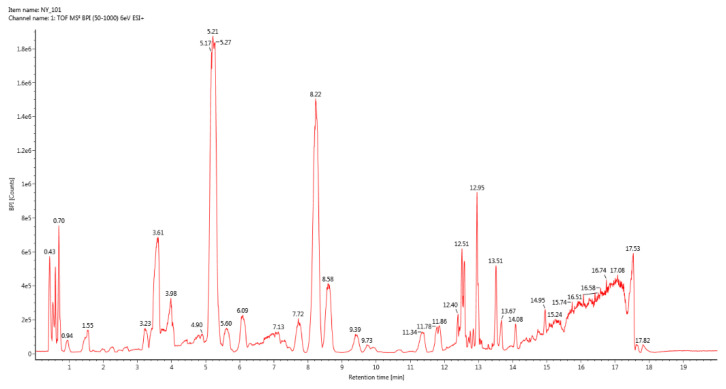
BPI total ion chromatogram of group 1.

**Figure 2 vetsci-09-00637-f002:**
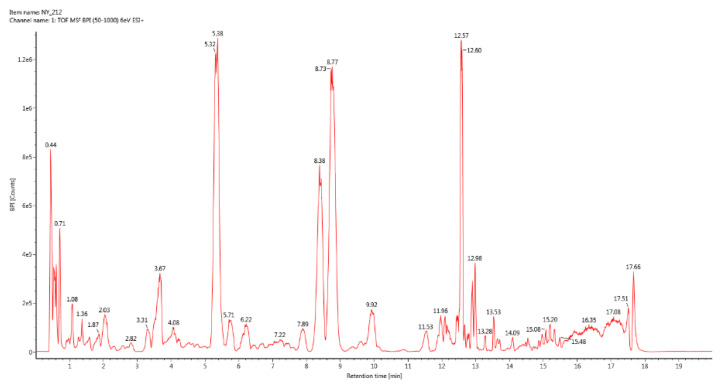
BPI total ion chromatogram of group 2.

**Figure 3 vetsci-09-00637-f003:**
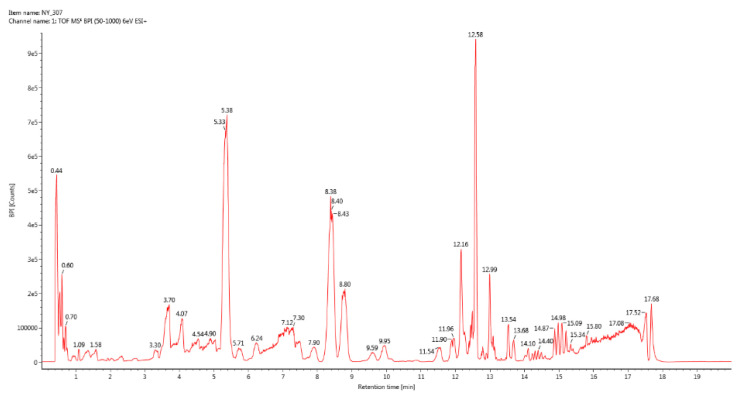
BPI total ion chromatogram of group 3.

**Figure 4 vetsci-09-00637-f004:**
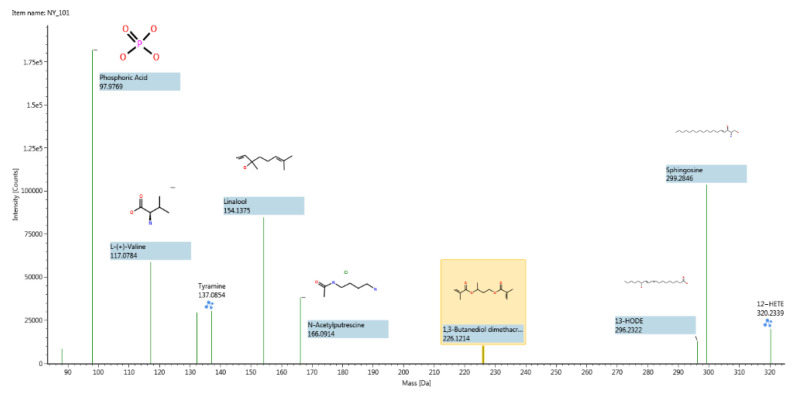
Structure determination of the major compounds in positive ion mode for Bactrian camels’ vaginal secretion samples.

**Figure 5 vetsci-09-00637-f005:**
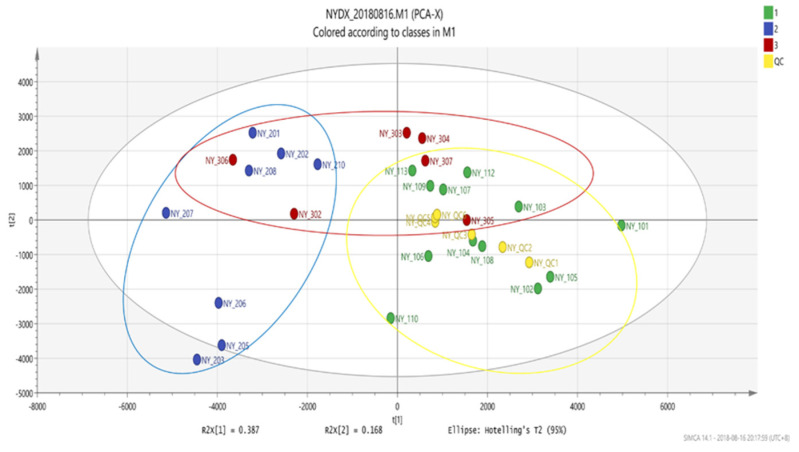
Score plot of PCA.

**Figure 6 vetsci-09-00637-f006:**
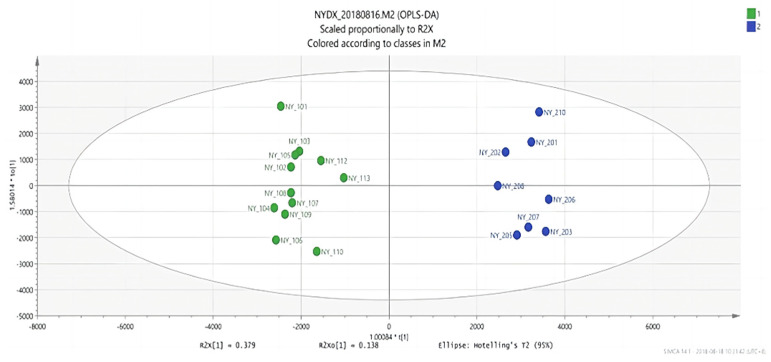
OPLS-DA score plots between group 1 and group 2.

**Figure 7 vetsci-09-00637-f007:**
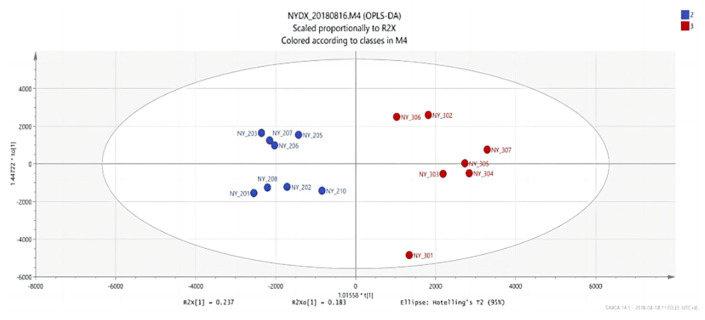
OPLS-DA score plots between group 2 and group 3.

**Figure 8 vetsci-09-00637-f008:**
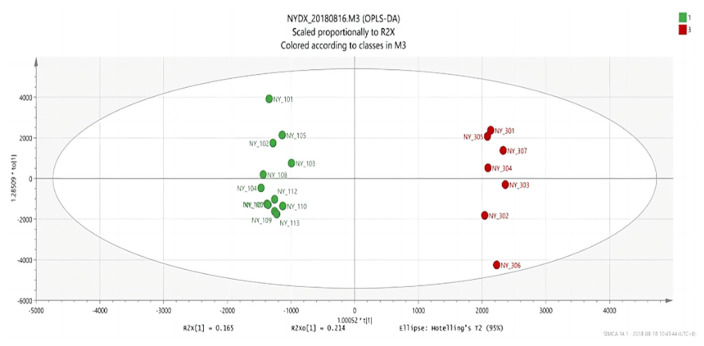
OPLS-DA score plots between group 1 and group 3.

**Figure 9 vetsci-09-00637-f009:**
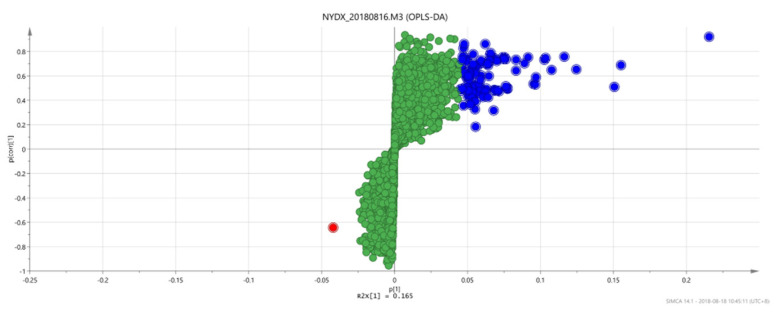
OPLS-DAS-plot diagram between group 1 and group 3. Note: Different colors indicate different correlations. The same below.

**Figure 10 vetsci-09-00637-f010:**
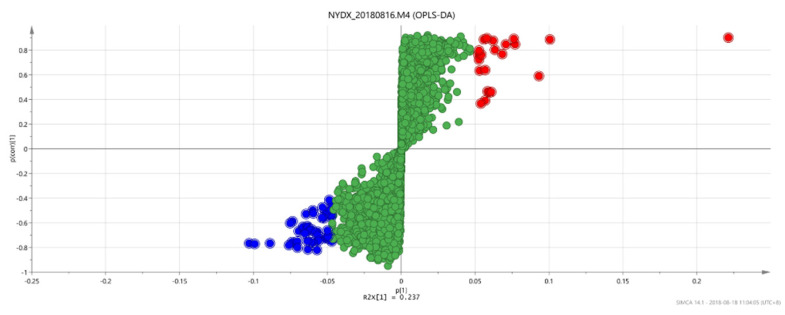
OPLS-DAS-plot diagram between group 2 and group 3.

**Figure 11 vetsci-09-00637-f011:**
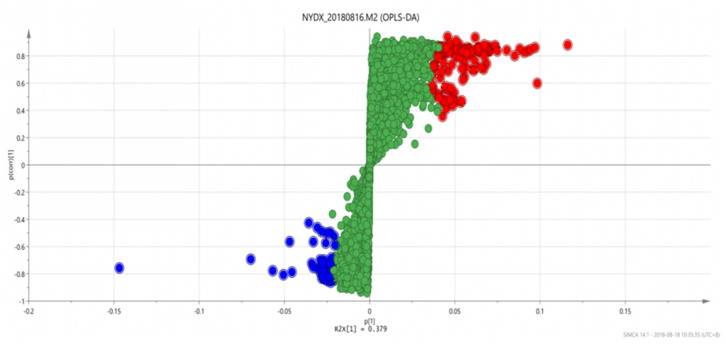
OPLS-DAS-plot diagram between group 1 and group 2.

**Figure 12 vetsci-09-00637-f012:**
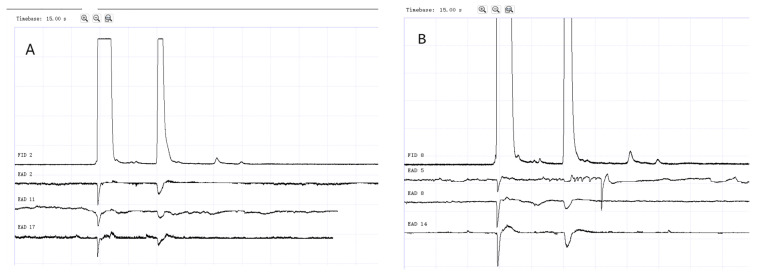
GC-EAD responses of 1-day-old *W. magnifica* to methylheptenone. (**A**) Male. (**B**) Female.

**Figure 13 vetsci-09-00637-f013:**
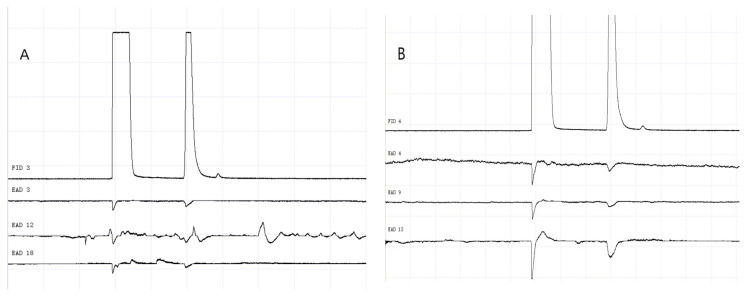
GC-EAD responses of 1-day-old *W. magnifica* to 1-octen-3-ol. (**A**) Male. (**B**) Female.

**Figure 14 vetsci-09-00637-f014:**
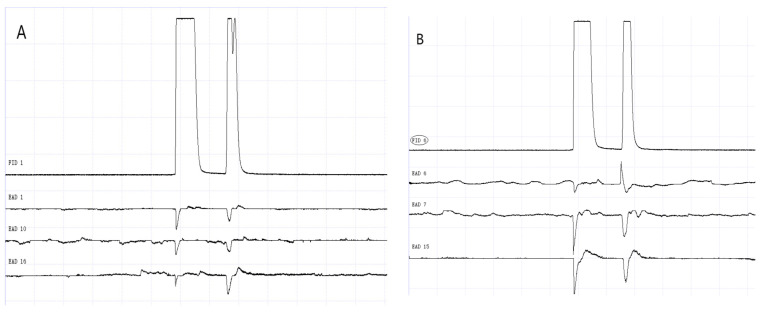
GC-EAD responses of 1-day-old *W. magnifica* to propyl butyrate. (**A**) Male. (**B**) Female.

**Figure 15 vetsci-09-00637-f015:**
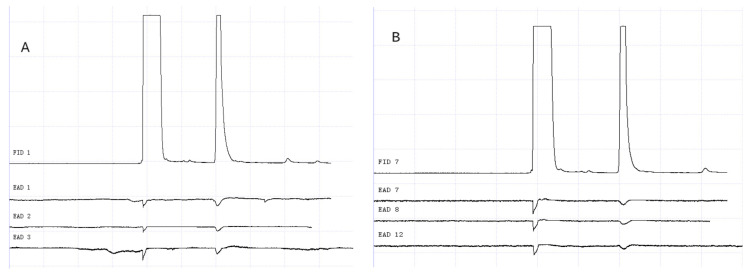
GC-EAD responses of 7-day-old *W. magnifica* to methylheptenone. (**A**) Male. (**B**) Female.

**Figure 16 vetsci-09-00637-f016:**
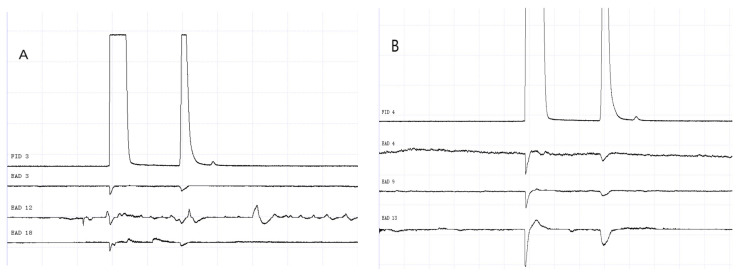
GC-EAD responses of 7-day-old *W. magnifica* to 1-octen-3-ol. (**A**) Male. (**B**) Female.

**Figure 17 vetsci-09-00637-f017:**
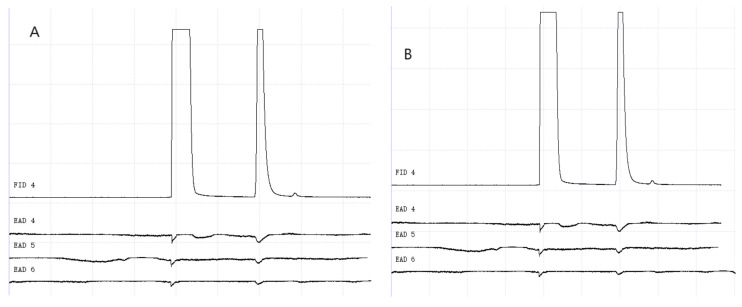
GC-EAD responses of 7-day-old *W. magnifica* to propyl butyrate. (**A**) Male. (**B**) Female.

**Figure 18 vetsci-09-00637-f018:**
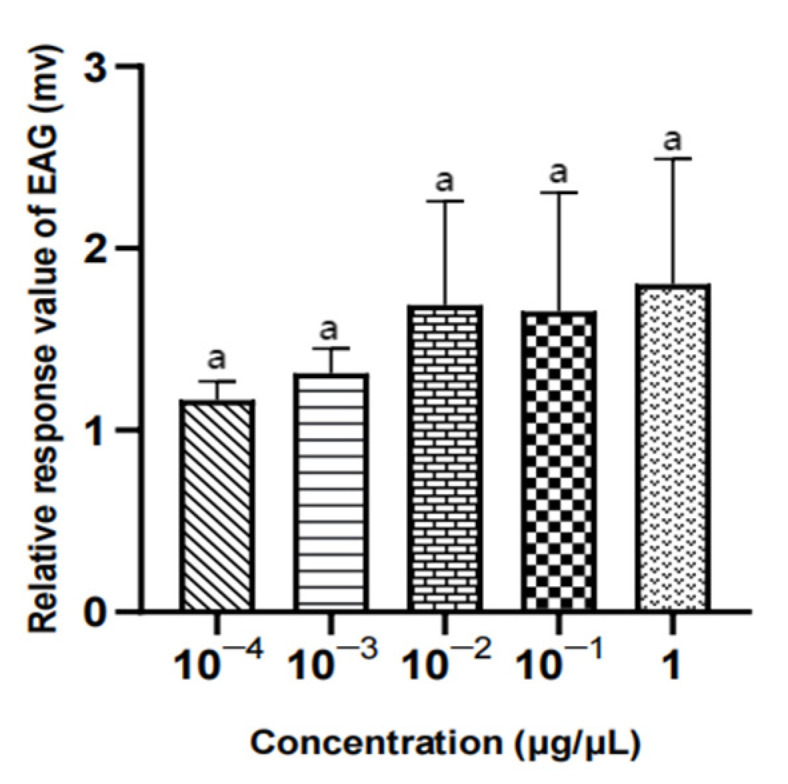
EAG responses of male *W. magnifica* to different concentrations of methylheptenone. Note: The same letter indicates no difference (*p* > 0.05), but different letters indicate significant difference (*p* < 0.05), so a and b are both indicate no difference, ab indicates significant difference. Different column shapes represent different concentrations of samples. The same below.

**Figure 19 vetsci-09-00637-f019:**
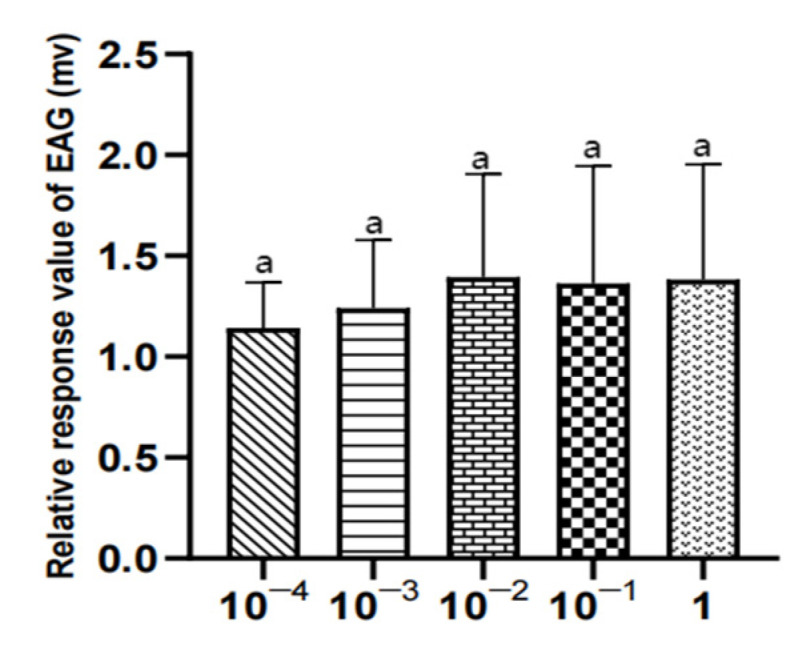
EAG responses of male *W. magnifica* to different concentrations of 1-octen-3-ol.

**Figure 20 vetsci-09-00637-f020:**
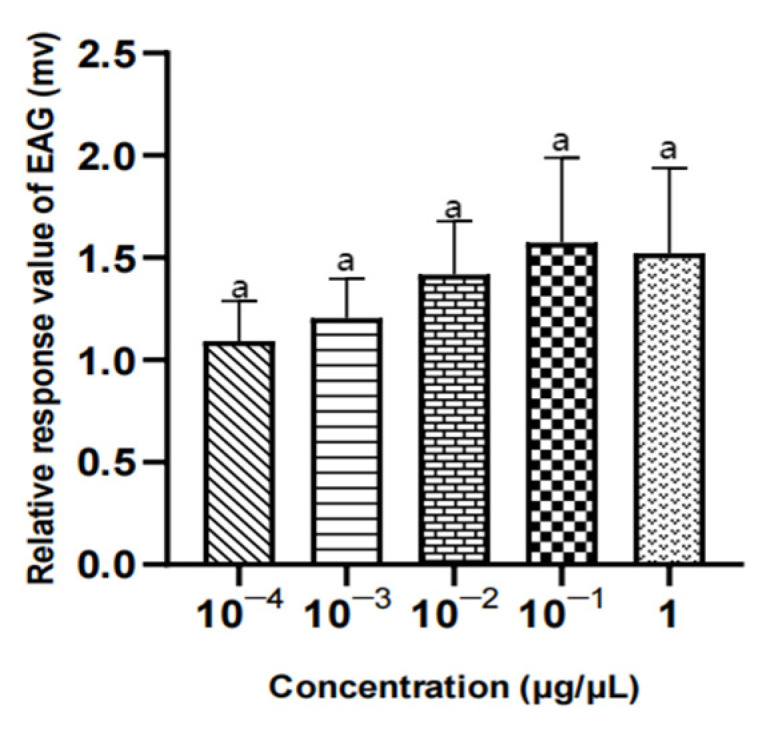
EAG responses of male *W. magnifica* to different concentrations of propyl butyrate.

**Figure 21 vetsci-09-00637-f021:**
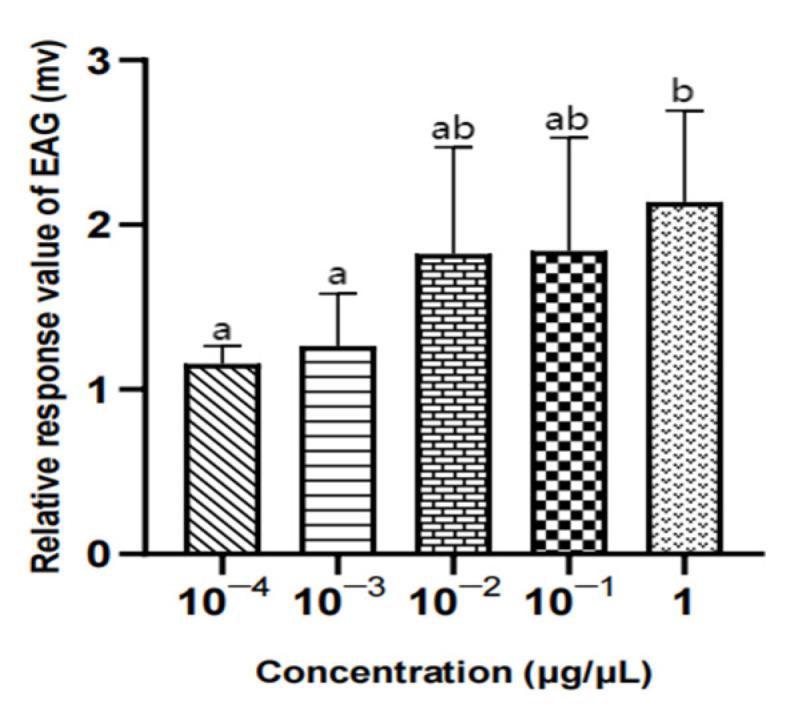
EAG responses of female *W. magnifica* to different concentrations of methylheptenone.

**Figure 22 vetsci-09-00637-f022:**
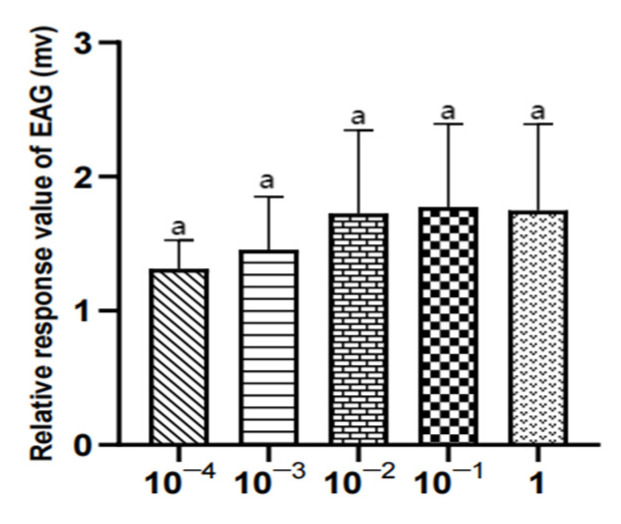
EAG responses of female *W. magnifica* to different concentrations of 1-octen-3-ol.

**Figure 23 vetsci-09-00637-f023:**
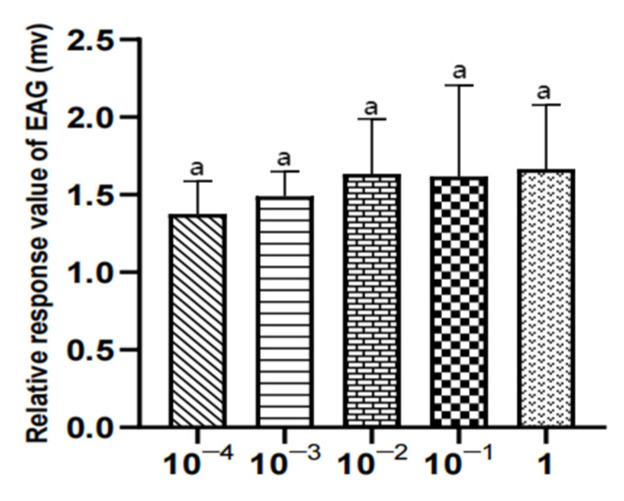
EAG responses of female *W. magnifica* to different concentrations of propyl butyrate.

**Figure 24 vetsci-09-00637-f024:**
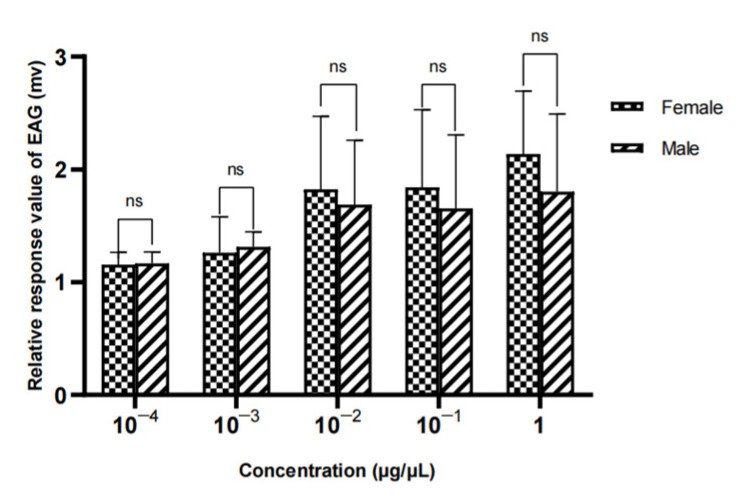
Comparison of EAG responses of *W. magnifica* to methylheptenone.

**Figure 25 vetsci-09-00637-f025:**
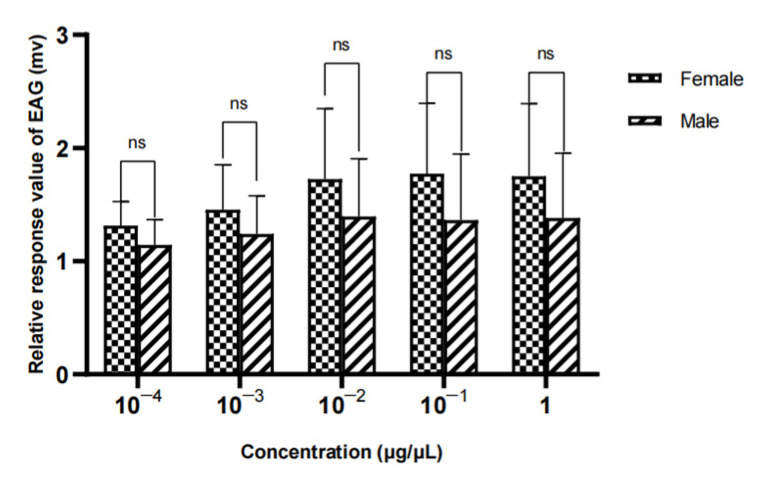
Comparison of EAG responses of *W. magnifica* to 1-octen-3-ol.

**Figure 26 vetsci-09-00637-f026:**
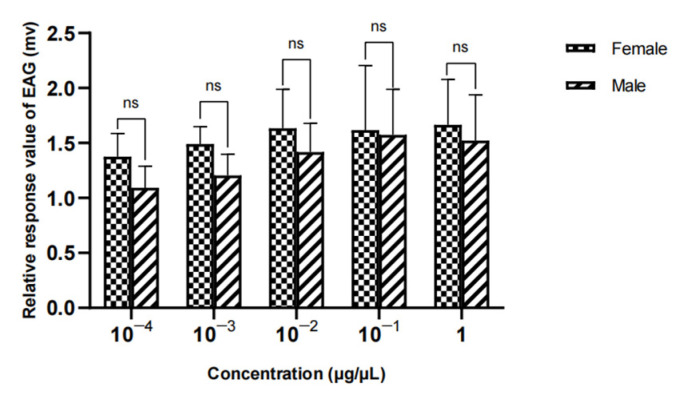
Comparison of EAG responses of *W. magnifica* to propyl butyrate.

**Figure 27 vetsci-09-00637-f027:**
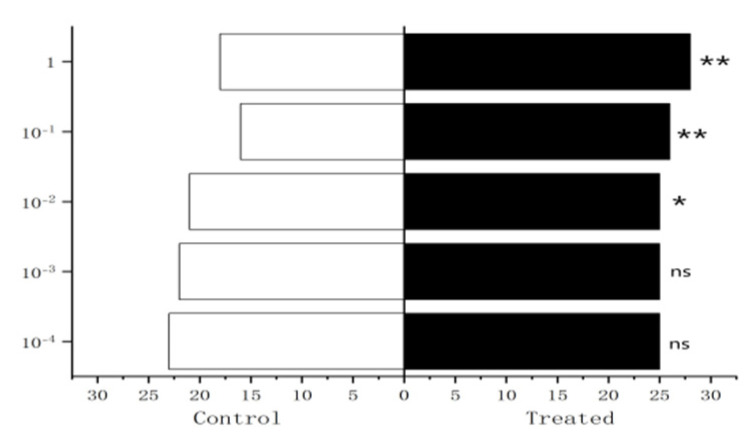
Behavioral responses to methylheptenone in male *W. magnifica*. Note: ns: no significant difference (*p* > 0.05); *: significant difference (0.01 < *p* < 0.05); **: extremely significant difference (*p* < 0.01); *x*-axis: sample grouping; *y*-axis: sample concentration. The same below.

**Figure 28 vetsci-09-00637-f028:**
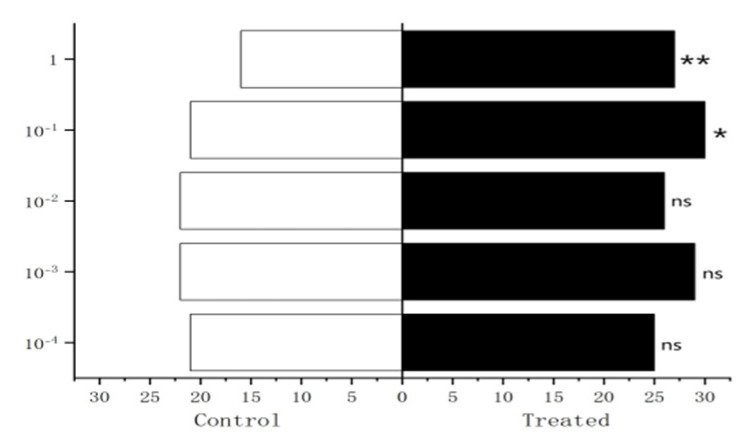
Behavioral responses to 1-octen-3-ol in male *W. magnifica*.

**Figure 29 vetsci-09-00637-f029:**
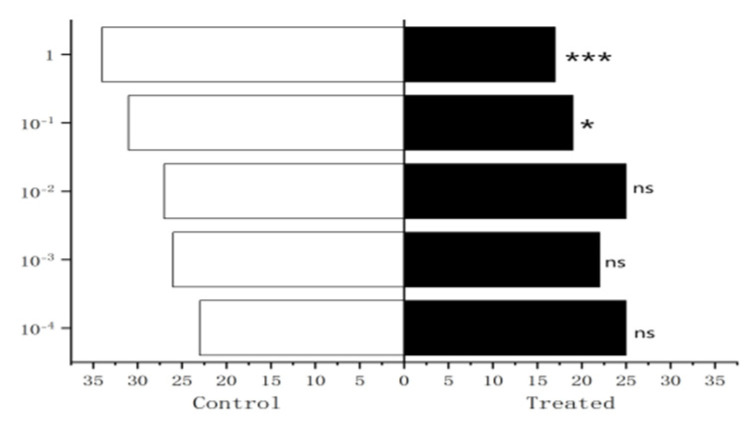
Behavioral responses to propyl butyrate in male *W. magnifica*.

**Figure 30 vetsci-09-00637-f030:**
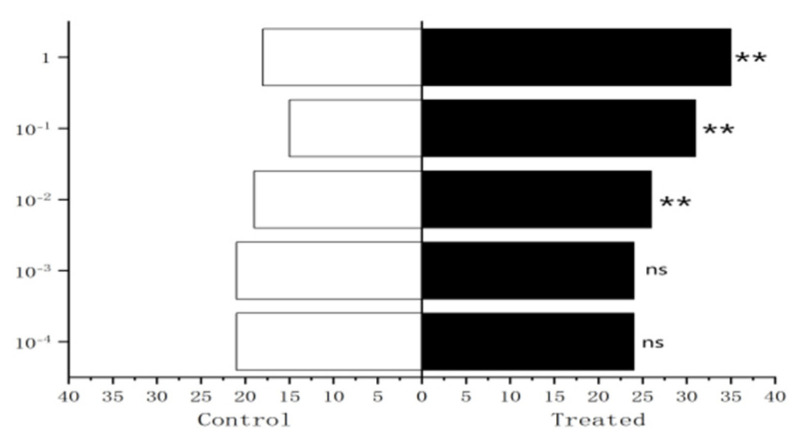
Behavioral responses to methylheptenone in female *W. magnifica*.

**Figure 31 vetsci-09-00637-f031:**
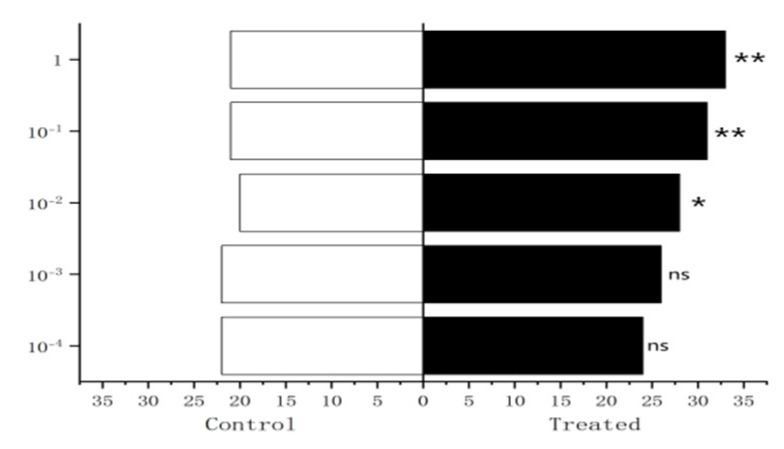
Behavioral responses to 1-octen-3-ol in female *W. magnifica*.

**Figure 32 vetsci-09-00637-f032:**
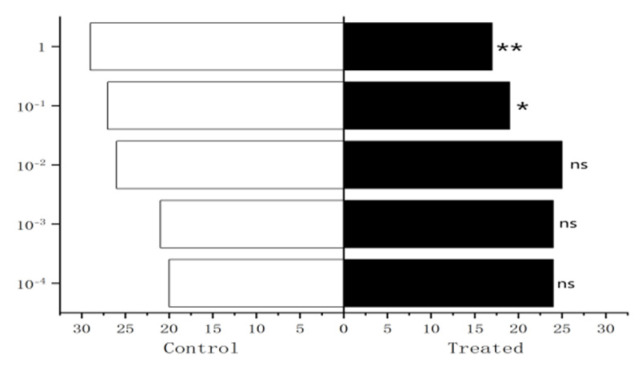
Behavioral responses to propyl butyrate in female *W. magnifica*.

**Figure 33 vetsci-09-00637-f033:**
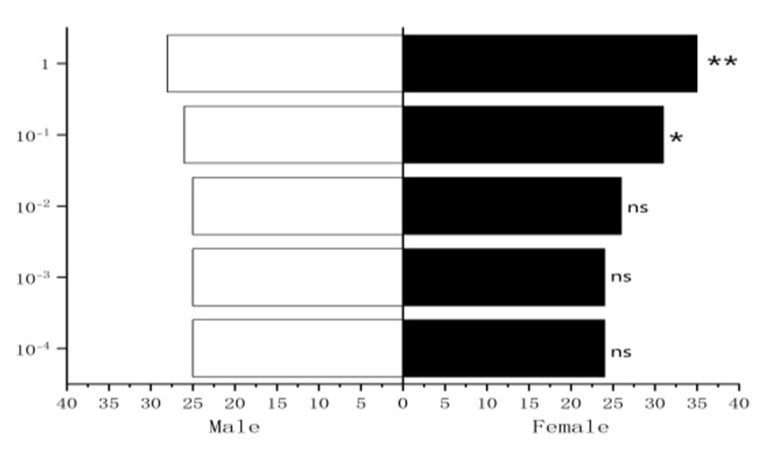
Comparison of behavioral responses of *W. magnifica* to methylheptenone.

**Figure 34 vetsci-09-00637-f034:**
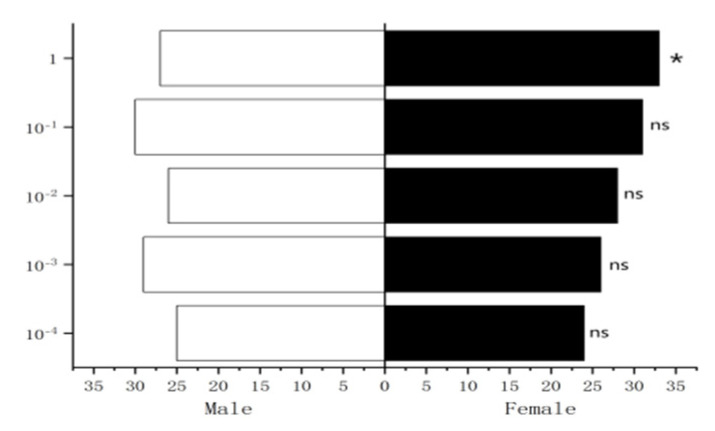
Comparison of behavioral responses of *W. magnifica* to 1-octen-3-ol.

**Figure 35 vetsci-09-00637-f035:**
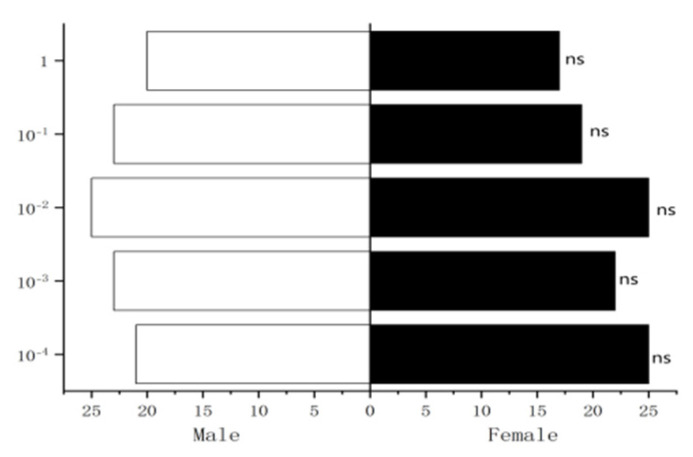
Comparison of behavioral responses of *W. magnifica* to propyl butyrate.

**Figure 36 vetsci-09-00637-f036:**
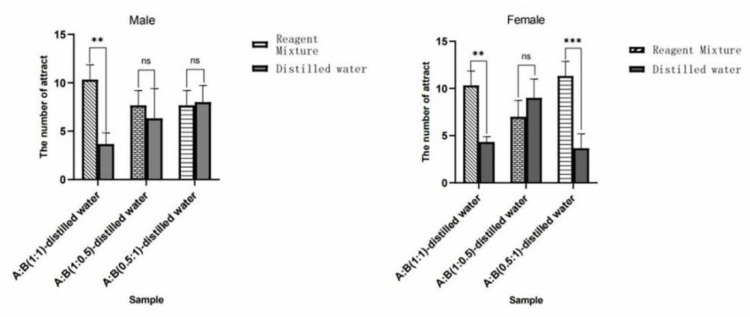
Behavioral responses of *W. magnifica* by mixing methylheptenone and 1-octen-3-ol in different proportions. Note: A: methylheptenone; B: 1-octen-3-ol; ns: no significant difference (*p* > 0.05); ** and ***: extremely significant difference (*p* < 0.01). The same below.

**Figure 37 vetsci-09-00637-f037:**
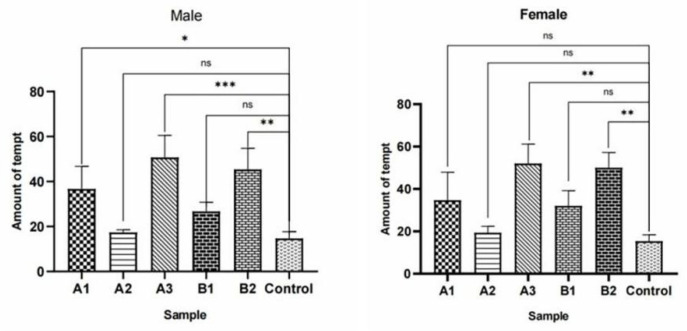
Attractive effects of different concentrations of methylheptenone and 1-octen-3-ol on *W. magnifica*. Note: A1: 10^−1^ µg/µL methylheptenone; A2: 10^−2^ µg/µL methylheptenone; A3: 1 µg/µL methylheptenone; B1: 10^−1^ µg/µL 1-octen-3-ol; B2: 1 µg/µL 1-octen-3-ol; ns: no significant difference (*p* > 0.05); *: significant difference (0.01 < *p* < 0.05); ** and ***: extremely significant difference (*p* < 0.01). The same below.

**Figure 38 vetsci-09-00637-f038:**
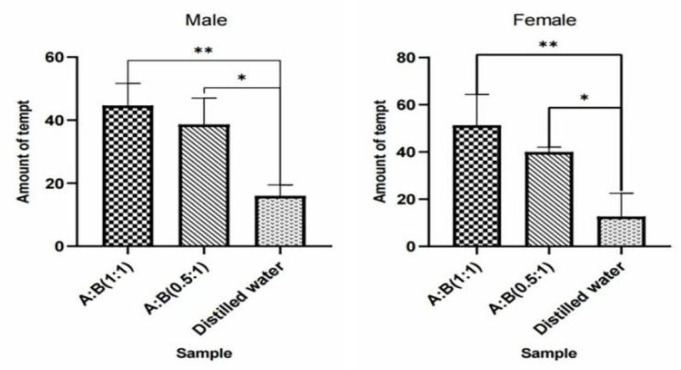
Attractive effects of *W. magnifica* by mixing methylheptenone and 1-octen-3-ol in different proportions.

**Table 1 vetsci-09-00637-t001:** Gradient mobile phase for Ultra Performance Liquid Chromatograph.

Time	Flow Rate (mL/min)	Mobile Phase A (%)	Mobile Phase B (%)
0	0.6	70	30
12	0.6	20	80
15	0.6	0	100
20	0.6	70	30

**Table 2 vetsci-09-00637-t002:** Mass spectrometry condition.

Project	Conditions
Collection quality range	50–1200 Da
Sweep time	0.1 s
Acquisition mode	ESI+, MSE
Lock mass	Leucine enkephalin (LE) 1 ppm (scanning time: 0.3 s; interval: 15 s)
A tube voltage	3 KV
Taper hole voltage	100 V
Collision energy (eV)	Low CE: 6/High CE: 20–50
Ionization source temperature	100 °C
Desolvation temperature	300 °C
Cone hole gas flow rate	100 L/h
Desolvation gas flow rate	800 L/h
Date acquisition time	20 min

**Table 3 vetsci-09-00637-t003:** The proportion of the mixture.

Reagent	Mixing Ratio
Methylheptenone:1-octen-3-ol	1:1
	1:0.5
	0.5:1

**Table 4 vetsci-09-00637-t004:** EAG relative response value of male *W. magnifica* to different concentrations of three compounds.

Sample Concentration	10^−4^ µg/µL	10^−3^ µg/µL	10^−2^ µg/µL	10^−1^ µg/µL	1 µg/µL
methylheptenone	1.167 mv	1.313 mv	1.688 mv	1.654 mv	1.803 mv
1-octen-3-ol	1.143 mv	1.240 mv	1.396 mv	1.362 mv	1.382 mv
Propyl butyrate	1.092 mv	1.204 mv	1.417 mv	1.573 mv	1.521 mv

**Table 5 vetsci-09-00637-t005:** EAG relative response value of female *W. magnifica* to different concentrations of three compounds.

Sample Concentration	10^−4^ µg/µL	10^−3^ µg/µL	10^−2^ µg/µL	10^−1^ µg/µL	1 µg/µL
methylheptenone	1.157 mv	1.263 mv	1.824 mv	1.841 mv	2.318 mv
1-octen-3-ol	1.313 mv	1.456 mv	1.726 mv	1.775 mv	1.749 mv
Propyl butyrate	1.375 mv	1.490 mv	1.635 mv	1.618 mv	1.666 mv

**Table 6 vetsci-09-00637-t006:** Statistical table of the trapping experiment for female *W. magnifica* with different concentrations of methylheptenone and 1-octen-3-ol.

Time			2 h	4 h	6 h	8 h	10 h	12 h
Concentration of Reagent (µg/µL)		Number of Experiments						
methylheptenone	10^−1^	1	2	8	20	30	30	38
		2	2	4	10	14	18	20
		3	4	6	12	24	34	46
	10^−2^	1	2	6	14	18	20	22
		2	0	2	2	10	12	16
		3	2	2	8	16	16	20
	1	1	4	8	16	24	34	44
		2	4	14	26	38	52	62
		3	2	2	8	16	16	20
1-octen-3-ol	10^−1^	1	0	4	10	20	24	26
		2	4	10	16	24	30	40
		3	2	8	10	14	24	30
	1	1	4	12	20	38	38	48
		2	8	16	28	40	50	58
		3	4	8	20	28	34	44
n-hexane		1	0	2	6	6	14	16
		2	2	4	4	4	8	18
		3	2	2	6	10	12	12

**Table 7 vetsci-09-00637-t007:** Statistical table of the trapping for male *W. magnifica* with different concentrations of methylheptenone and 1-octen-3-ol.

Time			2 h	4 h	6 h	8 h	10 h	12 h
Concentration of Reagent (µg/µL)		Number of Experiments						
methylheptenone	10^−1^	1	2	2	4	10	28	46
		2	2	6	12	16	18	26
		3	6	6	8	18	24	38
	10^−2^	1	4	6	10	16	18	18
		2	2	8	10	12	14	18
		3	0	2	2	6	8	16
	1	1	2	6	12	24	34	46
		2	2	10	20	30	44	62
		3	4	8	8	20	30	44
1-octen-3-ol	10^−1^	1	2	2	6	12	20	22
		2	2	6	10	16	26	28
		3	4	4	12	20	24	30
	1	1	0	6	8	18	28	38
		2	6	10	20	34	48	56
		3	2	8	10	24	34	42
n-hexane		1	0	2	6	8	10	14
		2	4	4	4	6	8	12
		3	0	2	6	6	12	18

**Table 8 vetsci-09-00637-t008:** Statistical table of the trapping for female *W. magnifica* by mixing methylheptenone and 1-octen-3-ol in different proportions.

Time		2 h	4 h	6 h	8 h	10 h	12 h
Mixing Ratio of Methylheptenone and 1-octen-3-ol	Number of Experiments						
1:1	1	4	4	12	28	52	64
	2	8	10	10	26	38	52
	3	2	6	8	8	24	38
0.5:1	1	4	8	8	15	27	40
	2	4	4	8	16	26	38
	3	8	10	10	20	28	42
Distilled water	1	8	8	12	20	24	24
	2	4	4	4	6	6	6
	3	0	2	2	6	8	8

**Table 9 vetsci-09-00637-t009:** Statistical table of the trapping for male *W. magnifica* by mixing methylheptenone and 1-octen-3-ol in different proportions.

Time		2 h	4 h	6 h	8 h	10 h	12 h
Mixing Ratio of Methylheptenone and 1-octen-3-ol	Number of Experiments						
1:1	1	4	4	12	20	30	44
	2	6	10	10	24	28	52
	3	4	6	16	26	34	38
0.5:1	1	4	8	16	20	44	48
	2	0	4	4	8	18	32
	3	4	8	10	16	24	36
Distilled water	1	12	12	20	20	20	20
	2	6	8	8	10	14	14
	3	0	4	4	6	12	14

## Data Availability

The datasets during and/or analyzed study available from the corresponding author and can be provided upon request.

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
