# Peer review of "Isolation and Identification of Volatile Substances with Attractive Effects on Wohlfahrtia magnifica from Vagina of Bactrian Camel"

_vetsci, 2022, doi:10.3390/vetsci9110637_

Round 1
Reviewer 1 Report
Line 55-56 «Human benefit from the insects, but many insects are also vectors of serious56 diseases in humans and animals»: not clear meaning to my understanding
Line 66: located”, change to locate
Line 77: Include full writing of the methods, like selected for gas chromatography-electroannetography(GC-EAD),31 eletroantennography (EAG), as it is the first reference of them in the main body of the paper text.
Line 106: what do you mean by “peaks”? samples maybe? They took 10 samples per animal of each group? Please
Author Response
Dear reviewer of Veterinary Sciences:
Thank you for your timely reply and constructive comments. I have revised the manuscript according to your comments.
- 《Human benefit from the insects, but many insects are also vectors of serious diseases in humans and animals》has been revised to《Some insects are good for humans, while others can cause serious diseases to humans and animals》.
- Located has been revisedto locate.
- The full names of GC-EAD and EAG have been added to the manuscript.
- Line 106 “peaks”has been revised to “samples”, it means group1, group2 and group3 collected 10, 10 and 6 samples respectively.
Looking forward to your reply.
Thank you for your support and good luck in your work!
Your sincerely,
Corresponding author:
Demtu Er
College of Veterinary Medicine, Inner Mongolia Agricultural University
Hohhot, 010018, P.R.China
E-mail: [email protected]
Reviewer 2 Report
Dear Authors, please correct the marked sentence. You should include all manufacturers which produced the used substances in your study. You should include the producer of your equipments which you used in your study too.
1. What is the main question addressed by the research?
The main question is to figure out that the methylheptenone and 1-octen-3-ol have significant attraction to W. magnifica.
2. Do you consider the topic original or relevant in the field? Does it address a specific gap in the field?
With these substances the W. magnifica can be trapped in the field to prevent vaginal myiasis in camels in the field.
3. What does it add to the subject area compared with other published material?
They used three substances for this study. They did not compare with commercial substances.
4. What specific improvements should the authors consider regarding the methodology? What further controls should be considered?
They could compare with commercial substances!
5. Are the conclusions consistent with the evidence and arguments presented and do they address the main question posed?
The conclusion is correct!
6. Are the references appropriate?
Yes
7. Please include any additional comments on the tables and figures.
The figures 27, ... y and x axis should be described

Author Response
Dear reviewer of Veterinary Sciences:
Thank you for your timely reply and constructive comments. I have revised the manuscript according to your comments.
- The marked sentence has been corrected.
- The manufacturers of the reagents and instruments involved in the study have been added to the manuscript.
- The x-axis and y-axis of figures 27 has been described in the manuscript.
Looking forward to your reply.
Thank you for your support and good luck in your work!
Your sincerely,
Corresponding author:
Demtu Er
College of Veterinary Medicine, Inner Mongolia Agricultural University
Hohhot, 010018, P.R.China
E-mail: [email protected]